# The Capacities of the Probiotic Strains *L. helveticus* MIMLh5 and *L. acidophilus* NCFM to Induce Th1-Stimulating Cytokines in Dendritic Cells Are Inversely Correlated with the Thickness of Their S-Layers

**DOI:** 10.3390/biom15071012

**Published:** 2025-07-14

**Authors:** Valentina Taverniti, Paolo D’Incecco, Stefano Farris, Peter Riber Jonsen, Helene Skovsted Eld, Juliane Sørensen, Laura Brunelli, Giacomo Mantegazza, Stefania Arioli, Diego Mora, Simone Guglielmetti, Hanne Frøkiær

**Affiliations:** 1Department of Food, Environmental and Nutritional Sciences (DeFENS), Division of Food Microbiology and Bioprocesses, Università degli Studi di Milano, 20121 Milano, Italy; laura.brunelli@unimi.it (L.B.); giacomo.mantegazza@unimi.it (G.M.); stefania.arioli@unimi.it (S.A.); diego.mora@unimi.it (D.M.); 2Sacco S.R.L., via Alessandro Manzoni 29/A, 22071 Cadorago, Italy; 3Department of Food, Environmental and Nutritional Sciences (DeFENS), Division of Food Sciences and Technology, Università degli Studi di Milano, 20121 Milano, Italy; paolo.dincecco@unimi.it (P.D.); stefano.farris@unimi.it (S.F.); 4Department of Veterinary and Animal Sciences, Copenhagen University, 1870 Frederiksberg, Denmark; peterriber@sund.ku.dk (P.R.J.); hms@chemometec.com (H.S.E.); juliane.s1993@gmail.com (J.S.); 5µbEat Lab, Department of Biotechnology and Biosciences (BtBs), University of Milano-Bicocca, Piazza della Scienza 4, 20126 Milano, Italy; simone.guglielmetti@unimib.it

**Keywords:** S-layer, thickness, Interleukin-12, endosomal degradation, transmission electron microscopy, atomic force microscopy

## Abstract

The two probiotic bacteria *Lactobacillus helveticus* MIMLh5 and *L. acidophilus* NCFM exhibit homology, are both equipped with an S-layer made up of highly homologous proteins and are capable of stimulating Th1-inducing signals in dendritic cells. In this study, we aimed to compare the two strains as regards the thickness of the S-layer and their capacity to induce the production of the two Th1-inducing cytokines IL-12 and IFN-β. For both bacteria, stimulation with an increasing number of bacteria led to the higher and prompter production of IL-12 and IFN-β, but at all MOIs tested, the IL-12 response induced by NCFM was always the strongest. For both bacteria, the induction of IL-12 peaked at a multiplicity of infection (MOI) of 2–5, while IL-10, known to inhibit the induction of IL-12 cytokines, was induced more slowly and continued to increase at a higher MOI. By employing specific inhibitors, MIMLh5 and NCFM were also shown to activate different MAP kinase pathways. Endocytosed MIMLh5 showed higher survival in the DCs compared to NCFM. In the presence of mannan, previously shown to accelerate endosomal killing of Gram-positive bacteria, the survival of MIMLh5 was strongly decreased, and IL-12 increased to a level close to that induced by NCFM without the addition of mannan, indicating the importance of rapid endosomal degradation for a strong IL-12 response. When measuring the S-layer thickness, MIMLh5’s S-layer appeared to be more than twice the thickness of NCFM and exhibited an elastic modulus approximately twice as high, which is a measure of a cell’s resistance to an applied mechanic stress. When the two strains were depleted of S-layer protein, the elastic modulus was comparable. Together, our data suggests that the thicker S-layer of MIMLh5 compared to NCFM may contribute to its endosomal survival, thus reducing its capacity to induce IL-12. This may constitute an important parameter in the selection of probiotic bacteria for specific purposes.

## 1. Introduction

The two species *Lactobacillus helveticus* and *Lactobacillus acidophilus*, even though they belong to and are adapted to different niches (dairy and gut environment, respectively), are phylogenetically strictly related (75% homology, with a 16S rRNA gene differing only by 1.6%; [1]). Major differences comprise the presence of insertion sequences (ISs) and the loss of host-interaction genes in *L. helveticus* [1]. Within these species, the probiotic strains *L. acidophilus* NCFM and *L. helveticus* MIMLh5 are coated with a proteinaceous surface layer (S-layer). The S-layer is a bi-dimensional crystalline array made of proteins covering the entire outer cell wall surface of several bacteria and archaea, which self-assembles through non-covalent interactions [2]. S-layer lattices have a highly porous structure and can present diverse symmetries and varying thickness (5–25 nm; [3]). Glycosylation has been described for species like *Lentilactobacillus buchneri*, *Len. kefiri*, *L. helveticus* and *L. acidophilus*. It is suggested that glycosylation is protective under challenging environmental conditions and important for cross-talk with specific immune receptors [3]. Recently, a group of S-layer-associated proteins (SLAPs) have been defined and demonstrated to affect the immune modulatory properties of bacteria [4,5,6,7,8]. The S-layer proteins of the *Lactobacillus* genus (in its previous widest classification, as referred to until 2020 [9]) are among the smallest described (25–71 kDa), have a high p*I* (9.35–10.4; [10]) and do not possess S-layer homology (SLH) domains binding to peptidoglycan-associated polymers [11]. The S-layers of *L. acidophilus* NCFM and *L. helveticus* MIMLh5 present a high percentage of homology (73% identity, 83% positivity; [12]), as also observed between NCFM and other *L. helveticus* strains [13]. The S-layer has been proposed and described as a bacterial molecular pattern involved in *L. acidophilus* NCFM and *L. helveticus* MIMLh5 interaction with host immune cells [14,15,16,17]. Indeed, in a previous study, we demonstrated that *L. helveticus* MIMLh5 S-layer protein promotes the uptake of the bacterium in dendritic cells [15], and other studies have likewise demonstrated that the S-layer can adhere to and interact with cells and specific surface receptors [18,19,20,21]. We have previously shown that the uptake of *L. acidophilus* NCFM is a prerequisite for strong interferon (IFN)-β and interleukin (IL)-12 in dendritic cells (DCs) [22,23].

Despite the above-mentioned similarities, *L. acidophilus* NCFM displays a much stronger ability to induce an IL-12 response in DCs than *L. helveticus* MIMLh5 [22,24,25]. IL-12 is essential for the activation of natural killer (NK) cells and differentiation of naïve T-cells to Th1 cells, with both cell populations being indispensable for an efficient elimination of bacterial infection and having allergy-counteracting capacity [26,27]. The strong Th1-stimulating ability of NCFM and other *L. acidophilus* strains is also a major reason for the widespread use of *L. acidophilus* in probiotic products [14,28,29]. In cell studies, *L. acidophilus* NCFM induces a potent IFN-β response, which in turn gives rise to a strong increase in IL-12, TNF-α and numerous anti-viral proteins [22]. The bacterium must be endocytosed and degraded endosomally to induce IFN-β, which consequently leads to an IL-12 response [14,15,16,17,18,19,20,21,22,24,25,26,27,28,29,30]. Upon isolating the S-layer protein from the two *Lactobacillus* strains, we noticed that the yield obtained from *L. helveticus* MIMLh5 was more than double the yield obtained from *L. acidophilus* NCFM. This may imply that the S-layer of *L. helveticus* MIMLh5 is thicker than that of *L. acidophilus* NCFM and thus confers better robustness to the bacteria against mechanical and chemical degradation. On this basis, we hypothesized that a slower endosomal degradation of the bacteria in myeloid antigen-presenting cells, such as DCs, leads to reduced production of IL-12. Accordingly, in this study we aimed to compare the cell surface features of the two *Lactobacillus* strains with a specific focus on the S-layer thickness by means of transmission electron and atomic force microscopy as well as assessing how readily they are degraded upon endocytosis in dendritic cells and their cytokine-inducing capacity.

## 2. Materials and Methods

**Bacterial strains, preparation and growth conditions**. *L. helveticus* MIMLh5, *L. acidophilus* NCFM and *L. delbrueckii* subsp. *bulgaricus* L92 (the latter used only in the atomic force microscopy assay as a negative control for the presence of the S-layer [31]) were grown in de Man–Rogosa–Sharpe (MRS) broth (Difco Laboratories Inc., Detroit, MI, USA). *Lactobacillus* strains were inoculated from frozen glycerol stocks and sub-cultured twice in MRS using a 1:100 (*v*/*v*) inoculum. To prepare fresh cultures to be used in immunological experiments, bacterial cells from an overnight culture were collected, washed twice with sterile PBS (Sigma-Aldrich, pH 7.4) and then resuspended in the same medium used to culture murine bone marrow-derived dendritic cells. To prepare PBS culture, bacterial cells from an overnight culture were collected, washed twice with sterile PBS, counted with the Neubauer Improved counting chamber, resuspended at 5 × 10^9^ cells mL^−1^ in PBS and stored at −80 °C in aliquots.

**Generation of bone marrow-derived dendritic cells**. Bone marrow-derived DCs were prepared as described previously [32]. Briefly, bone marrow from C57BL/6 mice (Tactonic, Lille Skensved, Denmark) was flushed out from the femur and tibia and washed. Bone marrow cells (3 × 10^6^ cells/mL) were seeded into Petri dishes of 10 mL aliquots in complete medium, consisting of RPMI 1640 (Sigma-Aldrich, St. Louis, MO, USA) containing 10% (*v*/*v*) heat-inactivated fetal calf serum supplemented with penicillin (100 U mL^−1^), streptomycin (100 mg mL^−1^), glutamine (4 mM) and 50 mM 2-mercaptoethanol (all purchased from Cambrex Bio Whittaker). The differentiation of bone marrow cells into DCs was enabled by the addition of 15 ng mL^−1^ murine granulocyte macrophage (GM) colony-stimulating factor (CSF) (harvested from a GM-CSF-transfected Ag8.653 myeloma cell line). The cells were incubated for 8 days at 37 °C in a 5% CO_2_ humidified atmosphere. On day 3, 10 mL of complete medium containing 15 ng mL^−1^ GM-CSF was added. On day 6, 8 mL was removed and replaced by 9 mL fresh medium. Non-adherent, immature DCs were harvested on day 8.

**Stimulation of DCs with bacteria**. Immature DCs (2 × 10^6^ cells mL^−1^) were resuspended in fresh medium, and 500 µL well^−1^ was seeded in 48-well tissue culture plates (Nunc, Roskilde, Denmark). *L. helveticus* MIMLh5 and *L. acidophilus* NCFM were added at a multiplicity of infection (MOI) of 1–50 to DCs and incubated at 37 °C in 5% CO_2_. In the MAPK inhibitor experiments, DCs were pre-incubated for 1 h with SP600125 (25 μM, JNK1/2 inhibitor, Invivogen, San Diego, CA, USA), SB203580 (10 μM, p38 MAPK inhibitor, Invivogen) and U0126 (10 μM, MEK1/2 inhibitor, Cell Signaling, Danvers, MA, USA). For the time-course experiments, DCs were harvested for RNA extraction after 2 h, 4 h, 6 h and 10 h; the supernatant was collected after 4 h, 6 h and 10 h. When DCs were pre-incubated with mannan (100 µg mL^−1^), 30 min prior to stimulation with bacteria (MOI 5), cells (2 × 10^6^ cells mL^−1^) were seeded into 48-well tissue culture plates (500 µL well) and incubated for 20 h before cytokine production was measured. When inhibiting IL-10’s action, DCs were seeded and 2 µg mL^−1^ of anti-mouse IL-10 R antibody was added (R&D systems, Minneapolis, MN, USA) 4 h after bacterial stimulation.

**Cytokine quantification in cell supernatant.** The concentration of murine IL-12 (p70), IL-10, IFN-β, TNF-α and IL-1β was analyzed by a sandwich enzyme-linked immunosorbent assay (ELISA) using Duoset^TM^ ELISA kits (R&D systems, Minneapolis, MN, USA).

**RNA extraction**. Stimulated DCs were harvested at different time points and total RNA was extracted using the MagMAX sample separation system (Applied Biosystems, Foster City, CA, USA), including a DNase treatment step for genomic DNA removal. RNA concentration was determined by Nanodrop (Thermo, Wilmington, DE, USA).

**Isolation of RNA, reverse transcription and quantitative PCR (RT-qPCR).** Five hundred nanograms of total RNA was reverse-transcribed by the TaqMan Reverse Transcription Reagent kit (Applied Biosystems, Foster City, CA, USA) using random hexamer primers according to the manufacturer’s instructions. The obtained cDNA was stored in aliquots at −80 °C. Primers and probes were obtained as previously described [30,33]. The amplifications were carried out in a total volume of 10 μL containing 1 × TaqMan Universal PCR Master Mix (Applied Biosystems), forward and reverse primer, TaqMan MGB probe and purified target cDNA (6 ng in reaction). The cycling parameters were initiated for 20 s at 95 °C, followed by 40 cycles of 3 s at 95 °C and 30 s at 60 °C, using the ABI Prism 7500 (Applied Biosystems). Amplification reactions were performed in triplicate, and DNA contamination controls were included. The amplifications were normalized to the expression of the β-actin-encoding gene. Relative transcript levels were calculated by applying the 2^(−ΔΔCT)^ method described by Livak and Schmittgen [34].

**Preparation of fluorescence-labeled bacteria.** For endocytosis experiments, *L. helveticus* MIMLh5 cells and *L. acidophilus* NCFM cells were fluorescently labeled using Alexa Fluor-conjugated succinimidyl-esters (SE-AF647; Alexa Fluor 647, Molecular Probes, Eugene, OR, USA). Bacterial cells in Dulbecco’s PBS (DPBS) were centrifuged for 5 min at 13,000× *g* in 1.5 mL Eppendorf tubes and resuspended in 750 μL of sodium carbonate buffer (pH 8.5); then SE-AF647 was added (10 µL for approximately 2 × 10^9^ bacterial cells mL^−1^). Bacteria were incubated at room temperature with agitation for 1 h in the dark, washed three times in sodium carbonate buffer and finally resuspended in the original volume of DPBS.

**Evaluation of bacteria uptake by DCs.** Non-adherent immature DCs were harvested and resuspended in complete medium to a cell density of 2 × 10^6^ cells mL^−1^. Pretreatment of cells with cytochalasin D (Sigma-Aldrich) at a final concentration of 0.5 µg mL^−1^ was performed in flasks for 1 h at 37 °C and 5% CO_2_ in a humidified atmosphere before the addition of stimuli. After that, 150 μL of DC suspension was seeded (3 × 10^5^ cells well^−1^) in 96-well U-bottom tissue culture plates (NUNC) and incubated for 30 min with fluorescence-labeled *L. helveticus* MIMLh5 or *L. acidophilus* NCFM (MOI 5), at 37 °C and 5% CO_2_ in a humidified atmosphere. All conditions were tested in triplicate, in at least three different experiments. After incubation with bacteria, DCs were spun down (400× *g* 5 min at 4 °C) and washed twice with cold PBS containing 1% FCS (washing buffer) and then fixed in washing buffer supplemented with 1% formaldehyde. Cells were analyzed by flow cytometry on a FacsCantoII (both from BD Biosciences). If not otherwise stated, the depicted data are representative of at least three independent experiments. Data analysis was performed using FLOWJO version 10 software (Ashland, OR, USA).

**Determination of bacterial survival in DCs.** Overnight cultures of *L. helveticus* MIMLh5 and *L. acidophilus* NCFM grown in MRS were centrifuged and the pellet washed with sterile PBS. The bacterial suspensions in PBS were then counted by means of a Neubauer counting chamber and brought to a cell density of 1 × 10^9^ cells mL^−1^. An aliquot of this bacterial suspension was diluted and plated on MRS agar, as a reference for the viability status of MIMLh5 and NCFM. A total quantity of 1 × 10^8^ bacterial cells was then added to 2 × 10^6^ DCs (MOI 50) seeded in 6-well plates in a DC medium, which was prepared without the addition of penicillin and streptomycin. Mannan (Sigma-Aldrich) was assessed by adding it to DCs at a concentration of 100 μg mL^−1^ 30 min before the incubation with bacterial cells. Each condition was tested in duplicate. After an incubation of 1 h at 37 °C in 5% CO_2_, the internalization of bacteria by DCs was blocked by aspirating the supernatant and by washing the DC layer twice with cold PBS, while keeping plates on an ice tray. Complete DC medium containing 100 μg mL^−1^ of gentamycin (Sigma-Aldrich) was then added to DCs, and plates were incubated for 1 h at 37 °C in 5% CO_2,_ in order to allow the killing of residual bacteria attached to DCs’ surface. After that, medium was aspirated and DCs were washed three times with PBS to eliminate residual gentamycin. Bacteria sensitivity to the gentamycin treatment was confirmed by exposing MIMLh5 and NCFM to the gentamycin concentration used. From this step, the survival of MIMLh5 and NCFM was evaluated by collecting DCs at three different time points, namely, 0 h, 1 h and 2 h, corresponding, respectively, to the time of incubation after 100 μg mL^−1^ gentamycin treatment, and 1 h and 2 h of incubation upon the first gentamycin treatment. At each time point, DCs were collected and resuspended in 0.1% Triton X100 in PBS and left under mild agitation for 30 min at room temperature to allow the permeabilization of DC membranes. Bacteria sensitivity to Triton treatment was preliminary assessed by exposing MIMLh5 and NCFM to different Triton concentrations (0.2%, 0.1%, 0.02% and 0.01% *v*/*v*) for 15 or 30 min. The number of colony-forming units (CFU) mL^−1^ after dilution and plating was compared with the CFU mL^−1^ of the bacteria incubated for 15 or 30 min in the presence of PBS. For conditions corresponding to 1 h incubation and 2 h incubation after the treatment with 100 μg mL^−1^ gentamycin, complete media containing 50 μg mL^−1^ gentamycin was used, and incubation at 37 °C in 5% CO_2_ was conducted, respectively, for 1 h and 2 h. At the end of Triton treatment, serial dilutions were prepared, and plating was performed on MRS agar media, followed by incubation at 37 °C for 72 h in anaerobic jars (Oxoid Anaerogen, Thermo Fisher Scientific, Osaka, Japan).

**Transmission electron microscopy**. *L. helveticus* MIMLh5 and *L. acidophilus* NCFM were sampled before and after LiCl treatment and prepared for transmission electron microscopy (TEM) as shown by D’Incecco et al. [35]. After washing with sterile PBS, samples were fixed at 4 °C for 2 h with a solution containing 2% glutaraldehyde and 2% paraformaldehyde in 0.1 M Na cacodylate solution buffered at pH 7.2 (Agar Scientific, Stansted, UK). Fixed cells were washed with buffer solution and then suspended in 100 µL of low-temperature gelling agarose (2% *w*/*v* in water, melted at 35–40 °C) (VWR, Milan, Italy). The suspension was cut into 1 mm^3^ cubes when gelled. The cubes were further fixed for 1 h at 4 °C and post-fixed in osmium tetroxide (EMS, Hatfield, PA, USA) (1% in water, *w*/*v*) in the dark for 2 h. Samples were dehydrated in a series of ethanol solutions (25, 50, 75, 90, 95 or 100%) for 15 min and then embedded in a Spurr resin (EMS, Hatfield, PA, USA), followed by curing at 60 °C for 24 h. Sections, 90 nm thick, were cut and stained with uranyl acetate and lead citrate (EMS, Hatfield, PA, USA), both 0.2% in water (*w*/*v*), prior to observations. Untreated samples were observed using a Leo912ab transmission electron microscope (Zeiss, Germany) at 100 kV. However, samples after LiCl treatment were observed using a Talos L120C transmission electron microscope (FEI, USA) at 100 kV because the Leo912ab microscope was not available anymore. S-layer thickness was measured by using ImageJ software, version 1.54 (Research Services Branch, National Institute of Health and Medicine, USA). At least three grids from 2 different resin blocks were observed.

**Atomic force microscopy.** An aliquot (1 μL) of freshly prepared bacterial suspensions of MIMLh5, NCFM and L92 (all bacteria at a concentration of 1 × 10^5^ cells mL^–1^) was placed onto a highest-grade V1 AFM mica disc (Ø = 20 mm; mean roughness <2 nm) (Nanovision srl, Brugherio, Italy). L92 was used as a negative control since it lacks an S-layer protein (Hynönen & Palva, 2013), [31]. Cells were fixed by air-drying at 40 °C and 50% RH for 15 min in a constant climatic chamber (mod. HPP, Memmert GmbH, Schwabach, Germany) in order to prevent complete dehydration. Nanomechanical tests were carried out on a Tosca™ 400 AFM (Anton Paar Italia srl, Rivoli, Italy) in contact resonance amplitude imaging (CRAI) mode, which has been demonstrated to be suitable for quantitative nanomechanical characterization of elastic and viscoelastic materials with a broad range of elastic moduli (Marinello, La Storia, Mauriello, & Passeri, 2019, [36]). To this end, an Arrow-FMR-10 Force Modulation probe (Nanoworld, Neuchâtel, Switzerland) featuring a rectangular cantilever with a triangular free end and a tetrahedral tip with a typical height of 10–15 µm was used. Additionally, the tip radius of curvature is ~10 nm. The cantilever has a spring constant and resonance frequency of 2.8 N m^–1^ and 75 kHz, respectively. First, 10 × 10 μm^2^ images and force–distance curves were recorded at ten different locations on the bacteria surface using Tosca™ Control software (Anton Paar Italia srl, Rivoli, Italy). Second, Tosca™ Analysis software (Anton Paar Italia srl, Rivoli, Italy) was used to analyze the force–distance curves and calculate the relevant nanomechanical information, that is, the elastic (Young’s) modulus (Appendix A). To this end, approach curves were fitted to the Hertzian model, which assumes the indentation to be negligible in comparison to the sample thickness, so that the substrate does not influence the calculations [37]. Accordingly, the indentation depth was set at 2 nm.

**Statistical analysis**. Statistical calculations were performed using the software program GraphPad Prism, version 5. The significance of the results was analyzed by the unpaired heteroscedastic Student’s t test with a two-tailed distribution or one-way ANOVA with Welch’s test or a test for multiple comparisons, as indicated in the figure legends. Differences of *p* < 0.05 were considered to be significant. ****: *p* < 0.0001; ***: *p* < 0.001; **: *p* < 0.01; *: *p* < 0.05.

**Ethics statement**. All animals used as a source of bone marrow cells were housed under conditions approved by the Danish Animal Experiments Inspectorate (Forsøgdyrstilsynet), Ministry of Justice, Denmark, and experiments were carried out in accordance with the guidelines ‘The Council of Europe Convention European Treaty Series 123 for the Protection of Vertebrate Animals used for Experimental and other Scientific Purposes’. Since the animals were employed as sources of cells, and no live animals were used in experiments, no specific approval was required for this study. The animals used for this study are covered by the general facility approval for the Faculty of Health and Medical Sciences, University of Copenhagen.

## 3. Results

### 3.1. IL-12 Induction by L. acidophilus NCFM and L. helveticus MIMLh5 Both Peak at MOI 2–5 but Result in Different IL-12 Levels

First, we compared the IL-12 and IL-10 response in DCs to increasing doses (MOI 1–50) of the two bacteria. The IL-12 concentration in the culture media peaked at MOI 2–5 for NCFM and MOI 2 for MIMLh5, while the IL-10 concentration continued to rise with an increasing bacteria concentration for both bacteria until MOI 50 (Figure 1). Next, we compared the effect of MIMLh5 and NCFM (both at MOI 2) on the induction of the expression of *Il12*, *Il10*, *Tnfa* and *Il1b* and the subsequent production of the cytokines after 2 h, 4 h, 6 h and 10 h of stimulation in DCs (Figure 2). The two *Lactobacillus* strains showed similar effects in the induction of the pro-inflammatory cytokines TNF-α and IL-1β (Figure 2). Specifically, regarding TNF-α, when DCs were stimulated either with MIMLh5 or NCFM, *Tnfa* expression peaked at 4 h, followed by a rapid decrease (Figure 2A), and the maximum cytokine concentration for both strains was reached at 6 h, after which the cytokine levels remained stable (Figure 2B). IL-1β was mildly induced by the two bacterial strains, and the expression of *Il1b* dropped over time from 2 h (Figure 2). The *Il10* expression continued to increase over the 10 h time frame after bacterial stimulation (Figure 2A). The *Il12* expression differed markedly between the two bacteria at 6 h and 10 h, where NCFM induced a fold change twice as high for *Il12* compared to MIMLh5 and resulted in an IL-12 production double that of MIMLh5 after 10 h (Figure 2).

### 3.2. L. helveticus MIMLh5 and L. acidophilus NCFM Differently Activate MAPK Pathways

Based on the observation that NCFM is a more potent inducer of IL-12, we hypothesized that MIMLh5 and NCFM differently activate signaling pathways in DCs. Since the activation of the MAP kinase cascade is crucial for signal transduction in immune cells [38], we treated DCs with inhibitors for MAP kinases—JNK1/2, p38 and MEK1/2—before stimulation (Figure 3). Specifically, DCs were stimulated with MIMLh5 or NCFM (both MOI 5) after 1 h of pre-incubation with the inhibitors. MIMLh5-induced production of IL-12 was mildly reduced with JNK inhibition (−21%), unlike NCFM (−72%), while only MIMLh5-induced IL-12 was inhibited by p38 inhibition (−25%) and increased by MEK inhibition (+39%) (Figure 3A). For the IL-10 production induced by the two bacterial strains, there were no significant differences between strains in the effect of the three inhibitors (Figure 3B).

### 3.3. The Respective Effect of L. helveticus MIMLh5 and L. acidophilus NCFM on IL12 Production Correlates with the Ability to Induce IFN-β

We have previously shown for several species of lactobacilli that IL-12 levels in DCs correlate with IFN-β production [23]. Therefore, we compared the expression of *Ifnb* in DCs induced by MIMLh5 (MOI 2) and NCFM (MOI 2). At 4 h of stimulation, the induction of *Ifnb* was the same for the two bacterial strains, but at 6 h, *Ifnb* expression was more than doubled upon NCFM stimulation, while after MIMLh5 stimulation, the expression decreased (Figure 4A). These results were confirmed in the ELISA, as we found that the concentration of IFN-β for MIMLh5- and NCFM-stimulated DCs differed from 543 pg mL^−1^ to 857 pg mL^−1^ (6 h) and 793 pg mL^−1^ to 1800 pg mL^−1^ (10 h), respectively (Figure 4B). To determine whether using a higher MOI would reduce the difference in the induced *Ifnb* fold change, we stimulated DCs with MOI 50 of the two bacterial strains. Using MOI 50 led to larger differences in both the gene expression and protein level at 4 h, whereafter the expression induced by both bacteria dropped noticeably, resulting in fairly constant IFN-β levels in the harvested supernatants (Figure 4A,B). Compared to the stimulation with MOI 2, the induction of IL-12 and IL-10 upon stimulation with MOI 50 was more prompt, reaching maximum levels around 6 h. IL-10 is known to inhibit the production of IFN-β and IL-12 and is induced at a later time point. Accordingly, to inhibit the action of IL-10, an antibody against the IL-10 receptor (αIL-10R) was added 4 h after bacterial stimulation (MOI 2 and MOI 50) (Figure 4C,D). At MOI 2, inhibiting the binding of IL-10 led to higher IFN-β and IL-12 production for both bacterial strains, while at MOI 50, no changes in the IFN-β response were observed; however, an increased IL-12 response was still visible. This suggests that in order to induce a strong IFN-β and IL-12 response, the dendritic cells must take up and degrade the bacteria before emerging IL-10 (and perhaps other mechanisms) inhibits the induced cell signaling.

### 3.4. L. helveticus MIMLh5 Displays a Higher Survival Ability in DCs

Fluorescently labeled *L. helveticus* MIMLh5 and *L. acidophilus* NCFM were used to evaluate the degree of uptake by DCs. To demonstrate that the bacteria were endocytosed, we pretreated DCs for 1 h with cytochalasin D. The percentage of DCs internalizing bacteria was comparable for the two bacteria (Figure 5A). We then examined the hypothesis that the lower IL-12 response induced by *L. helveticus* MIMLh5 compared to *L. acidophilus* NCFM could be caused by different degrading rates in DC endosomes. To this end, we evaluated the survival rate of MIMLh5 and NCFM inside DCs. The recovery of live bacteria from DCs was carried out at 0 h, 1 h and 2 h after incubation with the bacteria and subsequent gentamycin treatment. The DCs were treated with Triton X100 (0.1%) for 30 min, thus permeabilizing DCs without affecting the viability of either strain. The viable bacterial count was set to 100% at the time point 0 h (i.e., right after gentamycin treatment was used to kill non-endocytosed bacteria attached to DC surface), and after 1 h of incubation, the viability of *L. acidophilus* NCFM was reduced by 80%, whereas *L. helveticus* MIMLh5 viability was reduced by 14% (Figure 5B). After 2 h of incubation, 25% of the initial number of MIMLh5 bacteria were still alive, while the percentage of viable NCFM was reduced to 1% (Figure 5B). The importance of endosomal degradation of the two bacteria for strong IL-12 induction was further confirmed by demonstrating that the inhibition of phagosomal NADPH oxidase activation by apocynin [39] or phagocytic alkalinization by the addition of NH_4_Cl reduced IL-12 induction while only NH_4_Cl addition efficiently reduced IL-10 production (Figure 5C). We have previously shown that mannan induces ROS formation and enhances bacterial degradation [11,40]. Pre-incubation of DCs with mannan prior to bacterial stimulation enhanced endosomal killing (Figure 6A,B). The effect was stronger for MIMLh5, showing a reduction in viability after 1 h of incubation of 14% for untreated cells compared to 79% for mannan-treated samples (Figure 6A). For NCFM-stimulated samples, the viability of cells after 1 h was reduced by 79% in untreated cells compared to 93% in mannan-treated cells (Figure 6B). After 2 h, the percentage of viable MIMLh5 dropped from 75% in untreated cells to 4% in the mannan-treated cells, while the reduction in viable NCFM remained around 1%.

Together, these results demonstrate that the bacterial strains differ in their resistance toward phagosomal degradation in dendritic cells, which may explain their diverse/disparate abilities to induce IL-12.

### 3.5. L. helveticus MIMLh5 Carries a Thicker S-Layer Compared to L. acidophilus NCFM

Cell ultrastructure was investigated by TEM of washed cell pellets (Figure 7A). The S-layer was particularly evident in *L. helveticus* MIMLh5. In fact, measurements showed a significantly (*p* < 0.001) thicker S-layer in MIMLh5 with respect to NCFM (11.6 ± 1.6 nm vs. 6.87 ± 1.1 nm; Figure 7B). Moreover, the MIMLh5 S-layer showed thicknesses up to ~25 nm along the S-layer with no periodicity in appearance (Figure 7Ac). The removal of the S-layer after treatment with LiCl was evident in both bacteria (Figure 7Ad,Ae).

### 3.6. Elastic Modulus Measurements by AFM Reveal Higher Stiffness of L. helveticus MIMLh5 Surface Compared to L. acidophilus NCFM

Because TEM analysis revealed a thicker S-layer for MIMLh5 than NCFM, we decided to perform a nanomechanical test to check whether the thickness of the S-layer can be linked to the resistance of the cell surface to compressive stress (Table 1, Appendix A). With this aim, we prepared bacterial cells for surface characterization by AFM, before and after LiCl treatment. Moreover, besides MIMLh5 and NCFM, we also included a strain of *L. delbrueckii* subsp. *bulgaricus*, which does not possess an S-layer (Hynönen & Palva, 2013), [31], to rule out that LiCl might have prevalently affected cell surface properties. MIMLh5 showed elastic modulus values significantly higher than NCFM (5.69 ± 0.12 vs. 2.88 ± 1.02, Table 1). In addition, the removal of the S-layer with LiCl washing more evidently affected MIMLh5 (1.11 ± 0.82 after LiCl) than NCFM (1.24 ± 0.50 after LiCl, which is lower than, though not significantly different from, the values before LiCl; Table 1). This indicates a lower capacity to bear mechanical stress (lower stiffness) in the absence of an S-layer. As expected, *L. delbrueckii* subsp. *bulgaricus* strain L92 did not show elastic modulus impairment due to treatment with LiCl (Table 1).

**Table 1 biomolecules-15-01012-t001:** Elastic modulus measured at the surface of the three bacteria tested in this work, untreated and treated with LiCl. MIMLh5: *L. helveticus* MIMLh5. NCFM: *L. acidophilus* NCFM. L92: *L. delbrueckii* subsp. *bulgaricus* L92.

Sample	Elastic Modulus (GPa)
MIMLh5	5.69 ± 0.12 *^a^*
MIMLh5_LiCl	1.11 ± 0.82 *^b^*
NCFM	2.88 ± 1.02 *^b^*
NCFM_LiCl	1.24 ± 0.50 *^b^*
L92	6.79 ± 3.70 *^a^*
L92_LiCl	6.44 ± 0.89 *^a^*

Elastic modulus values are reported as means ± standard deviations. GPa: gigapascal. Each experiment and analysis was carried out in triplicate (three different cells) on ten different points, and the results are reported as mean values accompanied by their standard deviations (SDs). To determine statistically significant differences among the means (*p* ≤ 0.05), a one-way analysis of variance (ANOVA) was performed using SPSS version 20 statistical software (SPSS Inc., Chicago, IL, USA), followed by Tukey’s test for post hoc comparisons. Different superscript small letters denote a significant difference.

## 4. Discussion

Reportedly, lactic acid bacteria may trigger very different immune responses depending on strain-related properties and on the bacterial load to which host cells are exposed [41,42,43]. Diversity among microbial strains in the interaction with the host immune system may depend on the availability of ligands for the stimulation of Toll-like receptors (TLRs) and NOD-like receptors, in turn affecting the type and strength of the induced cytokine response [30,44,45]. However, the resilience of the bacteria to phagosomal degradation may also play a key role in the induced cytokine response. Here, we investigated in detail the cytokine response, in particular, the Th1-inducing response, induced by two *Lactobacillus* strains possessing quite similar genomes, both equipped with an S-layer, and found that a thicker S-layer and higher surface stiffness were paralleled with a poorer induction of the Th1-inducing cytokines IL-12 and IFN-β. This finding may help in the understanding of which bacterial properties are related to a specific cytokine profile and, thus, which strains may be suitable for specific purposes.

The characterization of surface molecules has been proposed as an essential step in designing tailored probiotics for specific uses [46]. The two strains *L. acidophilus* NCFM and *L. helveticus* MIMLh5 possess immunomodulatory properties that have been demonstrated to, at least partly, depend on their S-layer proteins [14,15,16]. In the current study, we demonstrated that the two bacteria are equally endocytosed by DCs and that both require endosomal acidification and NADPH oxidase activity for IL-12 production, thus ruling out the possibility that the difference in cytokine production is caused by different efficiencies in uptake or different needs for endosomal degradation for the induction of IL-12.

In contrast, the two strains differed with respect to the MAP kinase signaling pathways that they employ. Whereas *L. acidophilus*-induced IL-12 production depended on the MAP kinase JNK, *L. helveticus*-induced IL-12 both decreased by the inhibition of p38 and JNK and increased by the inhibition of MEK, strongly indicating the activation of different signaling pathways by the two bacterial strains. We have previously shown that *L. acidophilus* NCFM upon endosomal degradation stimulates the expression of *Ifnb* through the activation of the transcription factors IRF3/5 and JNK with no involvement of other MAP kinases and that IL-12 production depends on the production of IFN-β [22,33]. As *L. helveticus* MIMLh5 was affected by all three MAP kinases, this strain may involve other signaling pathways, e.g., the direct stimulation of IL-12 through NF-κB activation.

Interestingly, we found that although the two strains differed in their ability to induce IL-12, they peaked around the same concentration (MOI 2–5), while IL-10 continued to increase with increasing bacterial concentration for both bacteria. When using a higher MOI, the peak induction of IL-12 and IFN-β did not increase but was reached faster and the expression levels of the two genes dropped at an earlier time point upon stimulation. IL-10 is known as a resolving cytokine abrogating cell signaling pathways involving MyD88 [47,48]. Here, we show that the abrogation of the action of IL-10 on the lactobacilli-stimulated DCs four hours after bacterial exposure had a clear effect on the response induced by the two bacteria at MOI 5. In contrast, only IL-12 was slightly affected in MOI 50-stimulated DCs. IL-10 is produced (after 4–6 h); hence, the abrogation of IL-10 binding to the IL-10 receptor causes a down-regulation of IFN-β and IL-12. Moreover, as IFN-β is induced prior to IL-12 [49], after stimulation with MOI 50, only an effect on IL-12 production was seen. This indicates the importance of not only the bacterial load but also how fast the bacteria are degraded in causing the release of ligands for TLR stimulation. Indeed, comparing how many bacteria survived upon endocytosis in the DCs clearly showed that *L. helveticus* survived the conditions in the endosomes at a much higher rate than *L. acidophilus* NCFM. This suggests that resilience toward (enzymatic) degradation and other chemical attacks influencing the rate of bacterial degradation and the release of TLR-stimulating ligands is an important parameter determining the production of TLR-stimulating ligands and the subsequent TLR stimulation in the cells. The recognition of bacteria induces fast and robust production of reactive oxygen species (ROS) in the phagosomal lumen through the activation NADPH oxidase 2 (Nox2) [50,51], which further leads to the activation of enzymes, in particular, proteases such as cathepsins [52] and other peptidases [53], and reactive nitrogen species in the cell [54,55]. The action of such enzymes is expected to be essential for the degradation of the S-layer and thus for efficient degradation of the bacteria. However, the induction of Nox2 activation and ROS production may be delayed if the bacteria resist the initial degradation and release of TLR-activating ligands.

By TEM, we measured a markedly thicker S-layer in *L. helveticus* MIMLh5, and after LiCl treatment, we could isolate a greater amount of the S-layer protein. Moreover, MIMLh5 showed a higher resistance to elastic deformation, while when they were depleted of the S-layer, no difference was seen between the two strains, pointing toward a higher robustness of MIMLh5 due to the thicker S-layer. Thus, an increased thickness of the S-layer better protects against physical deformation and degradation, but most likely also against enzymatic and other chemical degradation. This was substantiated by the fact that prestimulation with mannan reduced the proportion of endocytosed *L. helveticus* MIMLh5 that survived at the level seen for *L. acidophilus* NCFM. We have previously shown that the addition of mannan stimulates prompt/strong induction of ROS production through the mannose receptor [40,55]. Hence, these facts indicate the importance of the activity of NADH oxygenase for efficient endosomal degradation of the protecting S-layer. Our data further establishes that faster degradation results in an increased production of IL-12 and thus that the bacterial degradation and consequent release of TLR-stimulating ligands must take place within a certain time frame upon endocytosis. If not, various feedback mechanisms including the induction of IL-10 may hamper the production of Th1-inducing cytokines.

A certain robustness is required for a probiotic bacterium to survive the environment in the gastrointestinal tract as well as the conditions present under the production and storage of food, and the S-layer plays an important protective role as recently shown for *L. helveticus* 34.9 [4]. To the best of our knowledge, here, we provide for the first time evidence that resilience toward endosomal degradation plays a key role in lactobacilli’s capacity to induce potent IL-12 production in dendritic cells. A major weakness of this study is obviously that we did not apply S-layer mutants to show causality between S-layer thickness and cytokine induction. However, we believe that this study will pave the way for future studies where such S-layer protein mutants and SLAP mutants can be involved to further demonstrate the single molecule’s impact on endosomal degradation and cytokine response. We suggest that the thickness of the S-layer, as illustrated by the two strains *L. helveticus* MIMLh5 and *L. acidophilus* NCFM, in dendritic cells may contribute to the reduced capacity to induce a strong Th1-inducing IL-12 and IFN-β response. Such knowledge should be taken into account in the future development and selection of probiotic strains for specific uses, e.g., the prevention of allergic and infectious diseases.

## 5. Conclusions

In conclusion, we show that *L. acidophilus* NCFM, as compared to *L. helveticus* MIMLh5, induces a stronger production of the Th1-inducing cytokines IFN-β and IL-12. This difference is caused by a faster endosomal degradation of *L. acidophilus,* and we suggest that the thicker S-layer of *L. Helvetic* MIMLh5 decreased the rate of its endosomal degradation.

## Figures and Tables

**Figure 1 biomolecules-15-01012-f001:**
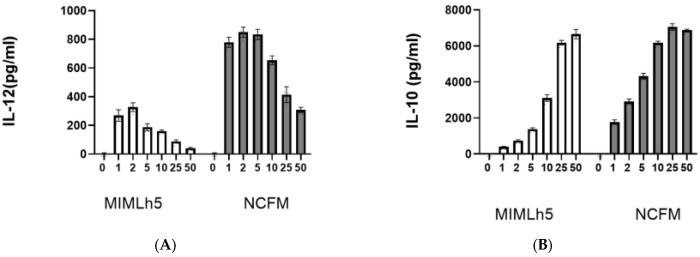
*L. acidophilus* is a stronger IL-12 inducer than *L. helveticus* MIMLh5. Cytokine production elicited in DCs by *L. helveticus* MIMLh5 and *L. acidophilus* NCFM at increasing ratios of bacteria to DCs (different multiplicity of infection, MOI 1–50). Concentration in supernatant harvested from DCs 20 h after stimulation with bacteria of IL-12 (**A**) and IL-10 (**B**).

**Figure 2 biomolecules-15-01012-f002:**
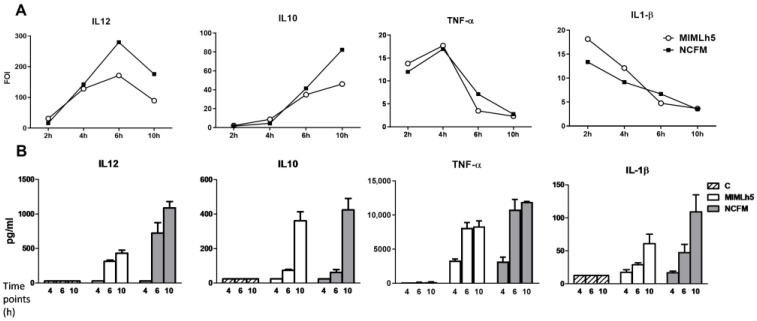
Fold change and cytokine production in DCs by *L. helveticus* MIMLh5 and *L. acidophilus* NCFM at multiple time points. (**A**): Gene expression of *Il12*, *Il10*, *Tnfa* and *Il1b* was determined by RT-qPCR after 2 h, 4 h, 6 h and 10 h incubation. Expression profiles are indicated as the fold change of induction (FOI) relative to unstimulated DCs, which was set at a value of 1. (**B**): Concentration of IL-12, IL-10, TNF-α and IL-1β in the supernatants of DCs as determined by ELISA after 4 h, 6 h and 10 h of incubation. *L. helveticus* MIMLh5 and *L. acidophilus* NCFM were both tested at MOI 2. Data presented are representative of three independent experiments. Data represent mean of measurements from triplicate cultures ± standard deviation and Welch’s ANOVA test.

**Figure 3 biomolecules-15-01012-f003:**
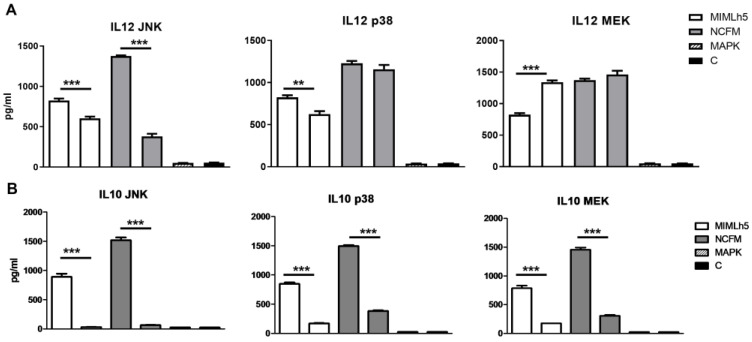
Different roles of MAP kinases in the induction of IL-12 by MIMLh5 and NCFM. Stimulation of DCs with *L. helveticus* MIMLh5 or *L. acidophilus* NCFM after pre-incubation with inhibitors for JNK, MEK and p38. Cytokine levels of IL-12 (**A**) and IL-10 (**B**) were measured in the supernatants of DCs by ELISA after 20 h. *L. helveticus* MIMLh5 and *L. acidophilus* NCFM were used at MOI 5. C: control (unstimulated DCs). JNK, MEK and p38: DCs added only with the respective MAPK inhibitors. Data presented are representative of three independent experiments. Data represent mean of measurements from triplicate cultures ± standard deviation and unpaired Student’s *t* test with two-tailed distribution. ***: *p* < 0.001.

**Figure 4 biomolecules-15-01012-f004:**
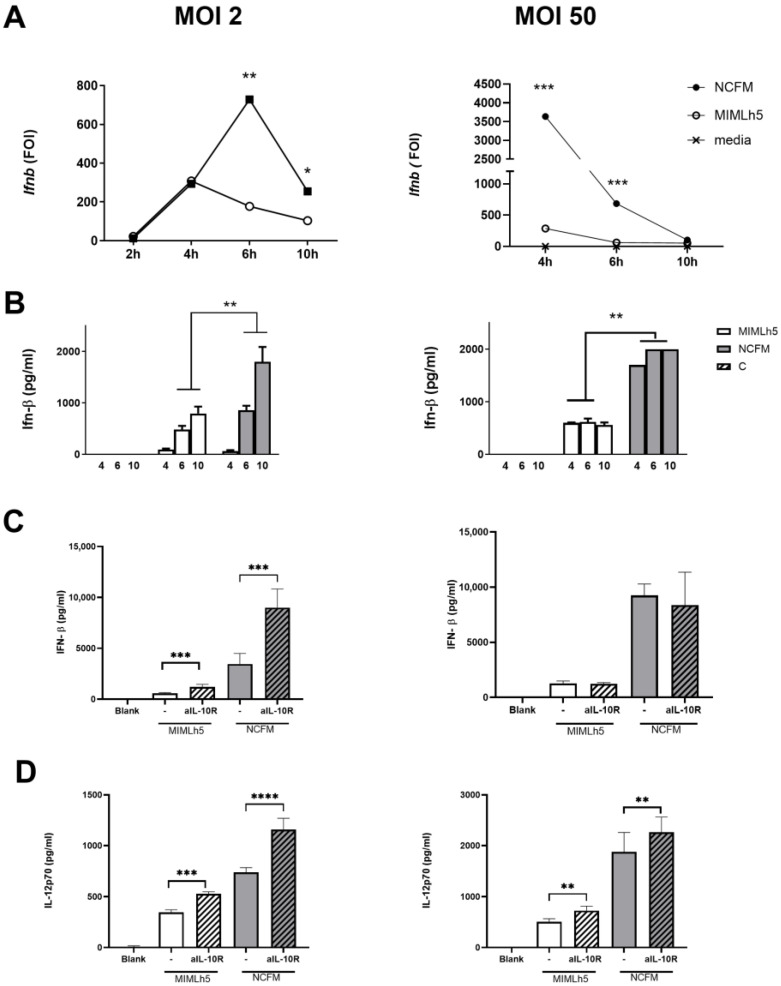
Interferon-β profile elicited in DCs by *L. helveticus* MIMLh5 and *L.acidophilus* NCFM at different time points. (**A**): Gene expression of *Ifnb* was determined by RT-qPCR after 2 h, 4 h, 6 h and 10 h incubation with bacteria (MOI 2 (left side) or MOI 50 (right side)). Expression profiles are indicated as the fold change of induction (FOI) relative to the control (unstimulated DCs), which was set at a value of 1. (**B**): Protein levels of IFN-β were measured in the supernatants of DCs by ELISA after 4 h, 6 h and 10 h of incubation. *L. helveticus* MIMLh5 and *L. acidophilus* NCFM were tested at MOI 2 (left side) and MOI 50 (right side). After 4 h of stimulation with either NCFM or MIMLh5, an anti-IL10 receptor antibody (αIL10R) was added to the cultures and IFN-β (**C**) and IL-12 (**D**) were measured 20 h after bacterial stimulation. Data presented are representative of at least two independent experiments. Data represents mean of measurements from triplicate cultures ± standard deviation and unpaired Welch’s ANOVA test. ****: *p* < 0.0001; ***: *p* < 0.001; **: *p* < 0.01, * *p* < 0.05.

**Figure 5 biomolecules-15-01012-f005:**
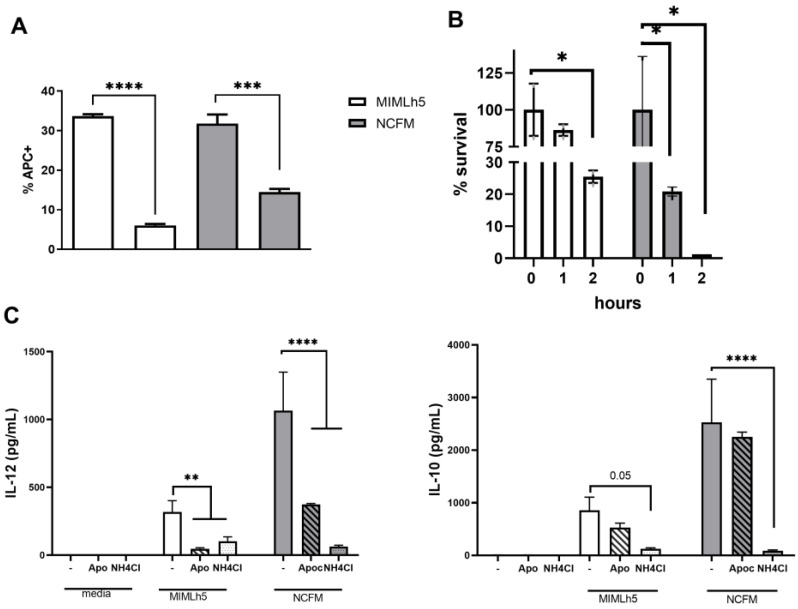
Endosomal degradation of *L*. *helveticus* MIMLh5 and *L. acidophilus* NCFM in DCs. (**A**): Flow cytometry analysis of DCs that had been pretreated with cytochalasin D (0.5 µg mL^−1^) or medium for 1 h before addition of Alexa Fluor 647-labeled *L. helveticus* MIMLh5 and *L. acidophilus* NCFM for 30 min. Data in the histograms report the percentage of the number of DCs positive for endocytosed MIMLh5 and NCFM with and without cytochalasin D treatment. Fluorescent bacterial cells were used at MOI 5. Allophycocyanin (APC)-positive population indicates DCs taking up bacteria. Data were compared by unpaired two-tailed *t* test. (**B**): Percentage of endocytosed bacteria in DCs at 0 h, 1 h and 2 h after bacterial uptake and subsequent gentamycin treatment to eliminate extracellular bacteria. Bacteria (MOI 50) were added to BMDCs and incubated for 1 h. After washing and gentamycin treatment, the number of endocytosed bacteria at 0 h, 1 h or 2 h was counted and the percentage survival calculated relative to 0 h of incubation. (**C**): IL-12 and IL-10 concentration in supernatants from DCs treated or not with apocynin or NH_4_Cl prior to stimulation with bacteria. Data were compared by ordinary one-way ANOVA with multiple comparisons; ****: *p* < 0.0001; ***: *p* < 0.001; **: *p* < 0.01; *: *p* < 0.05.

**Figure 6 biomolecules-15-01012-f006:**
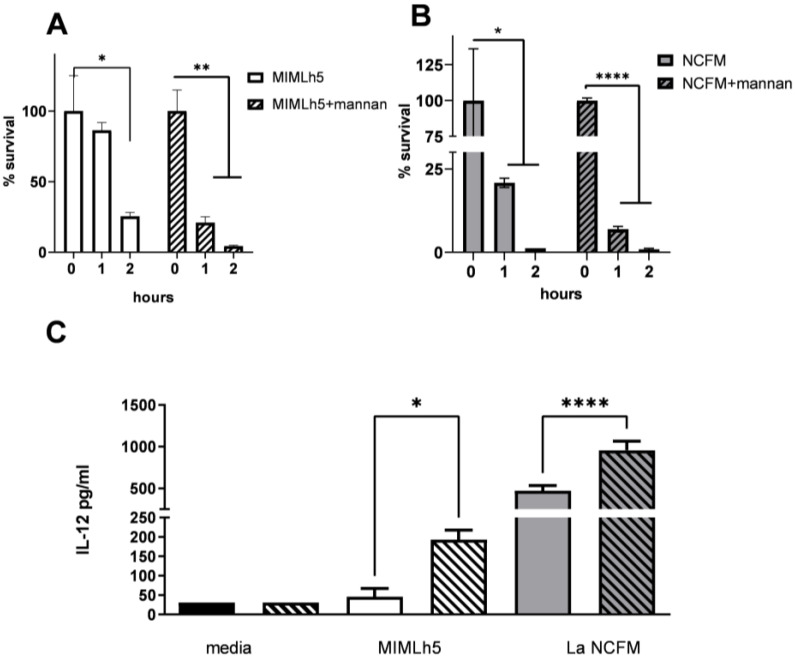
Mannan reduces endosomal survival of MIMLh5 and NCFM and enhances induced IL-12. Cells in presence or absence of mannan (100 µg mL^−1^) were incubated with bacteria (MOI 50) for 1 h. After washing and gentamycin treatment, the number of endocytosed bacteria at 0 h, 1 h or 2 h was counted and the percentage survival calculated relative to 0 h of incubation, (**A**): Cells incubated with MIMLh5; (**B**): cells incubated with NCFM. (**C**): IL12 concentration determined by ELISA in supernatant of DCs after 30 min pretreatment with mannan (100 µg mL^−1^) followed by 20 h incubation with bacterial cells. Black, white and gray bars: DCs added the indicated bacteria, hatched bars, DCs added mannan and the indicated bacteria. All data were compared by ordinary one-way ANOVA with multiple comparisons; ****: *p* < 0.0001; **: *p* < 0.01; *: *p* < 0.05.

**Figure 7 biomolecules-15-01012-f007:**
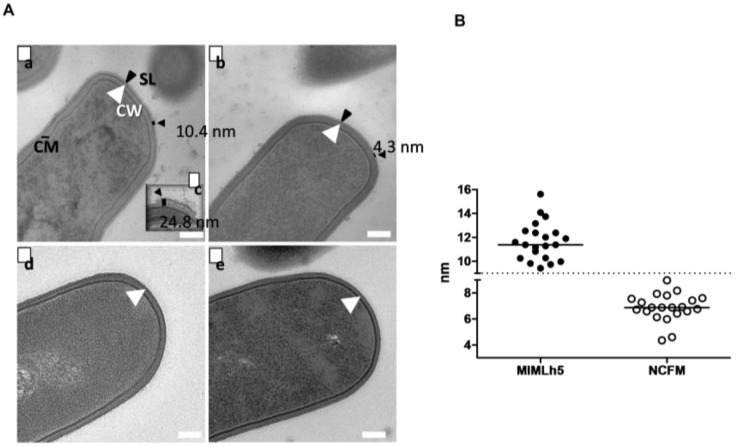
S-layer of MIMLh5 is thicker than that of NCFM. (A): Transmission electron microscopy images of ultra-thin sections of whole-cell preparation from *L. helveticus* MIMLh5 with thick S-layer (**Aa**) and *L. acidophilus* NCFM with thin S-layer (**Ab**). Panel (**Ac**) is an enlarged frame of a *L. helveticus* S-layer showing its further thickening. Panel (**A**(**d**,**e**)) show, respectively, MIMLh5 and NCFM after LiCl treatment. The black bar indicates the overall thickness of the S-layer. SL: S-layer (also indicated by black arrow); CW: cell wall (also indicated by white arrow). CM: cell membrane. White bar indicates a length of 100 nm. (**B**): Multiple determinations of the thickness of the S-layer (nm, *n* = 21) of the two bacteria.

## Data Availability

The original contributions presented in this study are included in the article/Appendix A. Further inquiries can be directed to the corresponding author.

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
