# Peer review of "The Capacities of the Probiotic Strains L. helveticus MIMLh5 and L. acidophilus NCFM to Induce Th1-Stimulating Cytokines in Dendritic Cells Are Inversely Correlated with the Thickness of Their S-Layers"

_biomolecules, 2025, doi:10.3390/biom15071012_

Round 1
Reviewer 1 Report
Comments and Suggestions for Authors
biomolecules-3675899-peer-review-v1
This study investigates the relationship between S-layer thickness in two probiotic Lactobacillus strains (L. helveticus MIMLh5 and L. acidophilus NCFM) and their capacity to induce Th1-associated cytokines (IL-12, IFN-β) in dendritic cells (DCs). The authors provide compelling evidence that the thicker S-layer of MIMLh5 correlates with reduced cytokine induction due to delayed endosomal degradation. The manuscript is well structured and clearly and understandably written. Hence, the MS may be published following a minor revision, according to the following comments:
- The description of statistical methods is vague. Specify whether ANOVA with post-hoc tests (e.g., Tukey’s) was used for multi-group comparisons.
- Add the statistical results in the figures.
- Line 438, where is Table 1?
- A representative force-distance curve (Fig. 7C) lacks axis labels.
- Address whether S-layer glycosylation (common in lactobacilli) contributes to structural resilience or immune evasion.
Author Response
1.The description of statistical methods is vague. Specify whether ANOVA with post-hoc tests (e.g., Tukey’s) was used for multi-group comparisons.
The post-tests used have now been specified in the methods and material section (l.271)
- Add the statistical results in the figures.
The statistics used has now been specified in methods, and the p-value intervals of relevance have been added in each figure caption.
- Line 438, where is Table 1?
Regrettably, Table 1 was by mistake not included in the submitted manuscript and is now included.
- A representative force-distance curve (Fig. 7C) lacks axis labels.
We are not sure if we understand this question. Figure 7 includes Fig 7A and 7 B
- Address whether S-layer glycosylation (common in lactobacilli) contributes to structural resilience or immune evasion.
Glycosylation has been described for several lactobacilli strains including Lentilactobacillus buchneri, Len. kefiri, L. helveticus and L. acidophilus.
We have included a sentence regarding this in the introduction (l.58).
- The description of statistical methods is vague. Specify whether ANOVA with post-hoc tests (e.g., Tukey’s) was used for multi-group comparisons.
Please refer to point 1 above.

Reviewer 2 Report
Comments and Suggestions for Authors
Generally, this research is of good quality, written by Authors with huge expertise on the topic. Through complex experiments (including transmission electron microscopy and atomic force microscopy) involving two Lactobacillus strains’ (L. helveticus MIMLh5 and L. acidophilus NCFM) and dendritic cells, they demonstrated that a thicker S-layer and higher surface stiffness was correlated with a lower induction of the Th1 stimulating cytokines IL-12 and IFN-beta. Essentially, the thicker S-layer of MIMLh5 (twice than that of NCFM) might have contribute to its endosomal survival and thus being able to reduce its capacity of inducing IL-12. Results appeared accurate and were presented in illustrative figures, with pertinent explanatory figure legends. I have listed my comments, for consideration, below:
1.Title: I would suggest to make it clearer - as it is somehow confusing about “revers capacities to induce”…”thickness of S-layers”. And why “reverse capacities”?. Only after reading the aim in the whole manuscript and especially the results, I realized what the title was about. Therefore, I was struggling to find a better alternative, like: “Capacities of probiotic strains L. helveticus MIMLh5 and L. acidophilus NCFM, to induce Th1 stimulating cytokines in dendritic cells, are inversely correlated with the thickness of S-layer” or anything else with the same meaning. From the original title, it was understood that ‘’revers capacities to induce” included also the thickness of S-layers, which did not make any sense.
2.Also, the Running title should be revised, to be clearer and to avoid redundancy of “S-layer”.
3.Abstract:
a.I encountered the same issue regarding the aim in the Abstract: “compared the two strains as regards their capacity to induce the production of the two Th1 inducing cytokines IL-12 and IFN-beta and the thickness of the S-layer.”. S-layer is not induced. Please revise.
b.Generally, maybe the Authors could kindly choose better what to present (and maybe enlarge) in the Abstract. After reading the whole manuscript, the Abstract is easily understandable. However, the Abstract should summarize the whole manuscript and be easily to be followed, before reading the full manuscript.
4.Introduction
a.Although this paragraph nicely introduces the topic and its relevance, there is still room for improvement, as most of references are quite old. This paragraph should be updated with recent references and corresponding data introduced in the text.
b.The aim by the end of Introduction is clearer (even though it requires polishing, including the English language). It should be summarized and used in Abstract. And, on this basis, the title should be reformulated (as I tried).
5.The methodology was described in details and it appears reproducible. Just please do not present Figures 7A and 7B before the previous numbers. Same for 7C and 7D. The figures have to be numbered in the order of their apparition in the text. There are no previous figures 1 to 6.
6.Results
a.The first paragraph – “L. acidophilus NCFM is a more potent IL-12 inducer than L. helveticus MIMLh5.” is not a new finding, as even the authors wrote in Introduction that “L. acidophilus NCFM displays a much stronger ability to induce interleukin (IL)-12 response in dendritic cells (DCs) than L. helveticus MIMLh5 (13-15).”
b.Writing style for all ILs should be the same everywhere. Please revise line 271 – “Il12, Il10, Tnfa and Il1b”, line 275, line 278 etc., including figure legends.
c.”L. helveticus MIMLh5 and L. acidophilus NCFM differently activate MAPK pathways.” In the text – lines 304-305, it was written “DCs were stimulated with MIMLh5 or NCFM (both MOI 2)”, while in figure legend 3 – line 314, it appears “L. helveticus MIMLh5 and L. acidophilus NCFM were used at MOI 5.”. Please revise and clarify.
d.Please revise lines 329-332.
e.“Moreover, besides MIMLh5 and NCFM we also included a strain of L. delbrueckii subsp. bulgaricus, which does not possess S-layer (Hynönen & Palva, 2013), to rule out that LiCl might have prevalently affected the cell surface properties.”. This should have been described in Methods, with reference inserted in the References list. Please do so.
f.From line 438 onwards (three times) – “Table 1” was written. However, there is no table in the manuscript. Please add, correct or do whatever is needed.
g.Figure 8 appears out of the blue, by the end of Results, without being mentioned anywhere in the text. This figure should be not placed there in any case, rather in Methods and, obviously, numbered correctly.
7.In Discussion, the Authors analyzed properly the findings of this research, and their meanings.
a.However, more could be added from the scientific literature, with updated references.
b.Also, I suggest deleting redundant data that were presented already in Introduction.
c.Although the strength is presented, the limitations are missing.
d.Also, proper directions for future research should be part of this paragraph.
7.Conclusion is not mandatory; however, it would be better to insert a conclusion before the last sentences of the paragraph.
8.References: at least 9 belong to the Authors – out of 41, representing 22%. I cannot say that there were inappropriate citations (as they represent the Authors’ work in this field), but as mentioned above, many more recent references should be used. Then, their own citation percentage will drop. Generally, all references are very old. We have 2 from 2020, 1 from 2021, 1 from 2022, nothing after.
9.I read attentively the iThenticate report and it looks fine (most considered data are in “MATERIALS AND METHODS”, which cannot be modified).
Comments on the Quality of English LanguageShould be improved.
Author Response
1.Title: I would suggest to make it clearer - as it is somehow confusing about “revers capacities to induce”…”thickness of S-layers”. And why “reverse capacities”?. Only after reading the aim in the whole manuscript and especially the results, I realized what the title was about. Therefore, I was struggling to find a better alternative, like: “Capacities of probiotic strains L. helveticus MIMLh5 and L. acidophilus NCFM, to induce Th1 stimulating cytokines in dendritic cells, are inversely correlated with the thickness of S-layer” or anything else with the same meaning. From the original title, it was understood that ‘’revers capacities to induce” included also the thickness of S-layers, which did not make any sense.
We thank this reviewer for the suggestion and have accordantly changed the title to: The capacities of the probiotic strains L. helveticus MIMLh5 and L. acidophilus NCFM to induce Th1 stimulating cytokines in dendritic cells are inversely correlated with the thickness of their S-layers.
2.Also, the Running title should be revised, to be clearer and to avoid redundancy of “S-layer”.
The running title is now deleted, as Biomolecules does not use running titles.
3.Abstract:
a.I encountered the same issue regarding the aim in the Abstract: “compared the two strains as regards their capacity to induce the production of the two Th1 inducing cytokines IL-12 and IFN-beta and the thickness of the S-layer.”. S-layer is not induced. Please revise.
The sentence has now been revised to: In this study, we aimed to compare the two strains as regards the thickness of the S-layer and their capacity to induce the production of the two Th1 inducing cytokines IL-12 and IFN-b (l.29)
b.Generally, maybe the Authors could kindly choose better what to present (and maybe enlarge) in the Abstract. After reading the whole manuscript, the Abstract is easily understandable. However, the Abstract should summarize the whole manuscript and be easily to be followed, before reading the full manuscript.
We have thoroughly revised the abstract and believe that it is now easier to follow.
4.Introduction
a.Although this paragraph nicely introduces the topic and its relevance, there is still room for improvement, as most of references are quite old. This paragraph should be updated with recent references and corresponding data introduced in the text.
New relevant references have now been added and described in the introduction.
b.The aim by the end of Introduction is clearer (even though it requires polishing, including the English language). It should be summarized and used in Abstract. And, on this basis, the title should be reformulated (as I tried).
We have reformulated the sentence (l.91).
5.The methodology was described in details and it appears reproducible. Just please do not present Figures 7A and 7B before the previous numbers. Same for 7C and 7D. The figures have to be numbered in the order of their apparition in the text. There are no previous figures 1 to 6.
References for figures in the method section have now been deleted.
6.Results
a.The first paragraph – “L. acidophilus NCFM is a more potent IL-12 inducer than L. helveticus MIMLh5.” is not a new finding, as even the authors wrote in Introduction that “L. acidophilus NCFM displays a much stronger ability to induce interleukin (IL)-12 response in dendritic cells (DCs) than L. helveticus MIMLh5 (13-15).”
This is correct. We have now changed the paragraph headline to: “IL-12 induction by L. acidophilus NCFM and L. helveticus MIMLh5 both peak at MOI 2-5 but result in different IL-12-levels.” In order to better reflect the new finding (l. 259).
b.Writing style for all ILs should be the same everywhere. Please revise line 271 – “Il12, Il10, Tnfa and Il1b”, line 275, line 278 etc., including figure legends.
We have corrected the writing style of the proteins. When we refer to the genes encoding the cytokines we use the writing style for murine genes, fx Il12)
c.”L. helveticus MIMLh5 and L. acidophilus NCFM differently activate MAPK pathways.” In the text – lines 304-305, it was written “DCs were stimulated with MIMLh5 or NCFM (both MOI 2)”, while in figure legend 3 – line 314, it appears “L. helveticus MIMLh5 and L. acidophilus NCFM were used at MOI 5.”. Please revise and clarify.
Thanks for noting this. MOI 5 is the correct quantity, and we have now corrected this in in line 304-305 (now l.282).
d.Please revise lines 329-332.
We have inserted table 1 and moved figure 8 to supplementary (now Supplementary Figure 1)
e.“Moreover, besides MIMLh5 and NCFM we also included a strain of L. delbrueckii subsp. bulgaricus, which does not possess S-layer (Hynönen & Palva, 2013), to rule out that LiCl might have prevalently affected the cell surface properties.”. This should have been described in Methods, with reference inserted in the References list. Please do so.
We have now included L.delbrueckii and reference in the Method description (l. 99 and l. 224).
f.From line 438 onwards (three times) – “Table 1” was written. However, there is no table in the manuscript. Please add, correct or do whatever is needed.
Table 1 was by mistake not included in the submission of the manuscript. This has now been corrected.
g.Figure 8 appears out of the blue, by the end of Results, without being mentioned anywhere in the text. This figure should be not placed there in any case, rather in Methods and, obviously, numbered correctly.
Figure 8 has been moved and is now Supplementary fig. 1
7.In Discussion, the Authors analyzed properly the findings of this research, and their meanings.
a.However, more could be added from the scientific literature, with updated references.
We have added some additional relevant references to the discussion
- Also, I suggest deleting redundant data that were presented already in Introduction.
We have now deleted in the discussion what we found was also described in introduction.
c.Although the strength is presented, the limitations are missing.
We have now emphasized the limitation of the study (l. 433)
d.Also, proper directions for future research should be part of this paragraph.
This is now included (l.439).
7.Conclusion is not mandatory; however, it would be better to insert a conclusion before the last sentences of the paragraph.
A conclusion paragraph has been inserted.
8.References: at least 9 belong to the Authors – out of 41, representing 22%. I cannot say that there were inappropriate citations (as they represent the Authors’ work in this field), but as mentioned above, many more recent references should be used. Then, their own citation percentage will drop. Generally, all references are very old. We have 2 from 2020, 1 from 2021, 1 from 2022, nothing after.
We have included a number of new references (from 2020 and onwards)
9.I read attentively the iThenticate report and it looks fine (most considered data are in “MATERIALS AND METHODS”, which cannot be modified).
We thank this reviewer for the very thorough and perceptive work, which has significantly improved our manuscript.

Reviewer 3 Report
Comments and Suggestions for Authors
The study entitled “The probiotic strains L. helveticus MIMLh5 and L. acidophilus 2 NCFM show reverse capacities to induce Th1 stimulating cyto- 3 kines in dendritic cells and thickness of S-layers” compares two strains of Lactobacillus in the ability to stimulate interleukin production when internalized in DCs and demonstrates that the differences in these effects are attributable to the survival time in turn dependent on the robustness of the S-layer and time of bacterial cell degradation in DCs. It is not clear why it is important to differentiate these bacteria based on IL-12 induction capacity so this must be explained in the introduction as also the implications of bacteria internalization in DCs should be commented in the introduction: is this an advantage for the probiotic action? Is this a common mechanisms for probiotics?. Discussion: please, reiterate less the results in favor of comments and possible explanations. While carrying out the revisions the authors could switch to the journal style by using the appropriate template and add the abbreviation list according to the instructions for authors. The English style should be re-checked.
Specific remarks:
Line 78: DCs
Line 87: which were the growth conditions?
Line 91: for what were PBS cultures used?
Line 103: please, spell out GM-CSF
Line 119: they were seeded
Line 127: RNA in a new line
Line 132: preparation of RNA?
Line 197, 500, 530: here and elsewhere: spacing
Line 274: In specific does not seems English
Line 279: after 2 h?
Lines 329-332: please, correct
Line 339: punctuation
Lines 344-345, 404, 421: please, check italics an abbreviations
Lines 372-376, 394-401: please, mention this in the Materials and methods section
Lines 405-408: this must not go in the figure legend
Line 427: please, correct the description of the white bar
Line 458: inducing…induced
Line 454, 499-506: you followed changes in interleukins and here you discuss about receptors (that you should spell out), please, make the connection explicit
Lines 506-509: please, give a sense to this statement in relation to this study
Author Response
The study entitled “The probiotic strains L. helveticus MIMLh5 and L. acidophilus 2 NCFM show reverse capacities to induce Th1 stimulating cyto- 3 kines in dendritic cells and thickness of S-layers” compares two strains of Lactobacillus in the ability to stimulate interleukin production when internalized in DCs and demonstrates that the differences in these effects are attributable to the survival time in turn dependent on the robustness of the S-layer and time of bacterial cell degradation in DCs.
It is not clear why it is important to differentiate these bacteria based on IL-12 induction capacity so this must be explained in the introduction
IL-12 production by dendritic cells is required to induce a Th1 cell response that protects against bacterial infections and counteract allergic responses. This is stated in the introduction (l. 76-80).
as also the implications of bacteria internalization in DCs should be commented in the introduction: is this an advantage for the probiotic action?
A sentence regarding the requirement for uptake of l acidophilus for the production of IL-12 has been added (l. 72-74).
Is this a common mechanisms for probiotics?.
Yes, at least for the Lactobacillus types. They have to be phagocytosed in order to induce cytokine production.
Discussion: please, reiterate less the results in favor of comments and possible explanations. While carrying out the revisions the authors could switch to the journal style by using the appropriate template and add the abbreviation list according to the instructions for authors. The English style should be re-checked.
We have revised the discussion, and believe that only where it is needed results are summarized.
Specific remarks:
Line 78: DCs has been corrected (l.89)
Line 87: which were the growth conditions? At 37C anaerobically.
Line 91: for what were PBS cultures used? These were the bacterial stocks used for the cell experiments described from l. 118
Line 103: please, spell out GM-CSF GM-CSF is now spelled out (l.113)
Line 119: they were seeded This has now been added (l.129)
Line 127: RNA in a new line corrected
Line 132: preparation of RNA? Has been cored to isolation (l.139)
Line 197, 500, 530: here and elsewhere: spacing. We have checked the manuscript thoroughly and corrected extra and missing spaces.
Line 274: In specific does not seems English. ‘In specific’ is an English term, but has now beencorrected to ‘Specifically’, which seems more correct in the used context .
Line 279: after 2 h? we did not include 2h
Lines 329-332: please, correct done.(l.345)
Line 339: punctuation corrected(l.356)
Lines 344-345, 404, 421: please, check italics an abbreviations. We have checked the entire manuscript and corrected accordingly.
Lines 372-376, 394-401: please, mention this in the Materials and methods section. This is mentioned in M&M (respectively, l.122 and l.202)
Lines 405-408: this must not go in the figure legend -has been changed.
Line 427: please, correct the description of the white bar -has been corrected (l.724)
Line 458: inducing…induced
Line 454, 499-506: you followed changes in interleukins and here you discuss about receptors (that you should spell out), please, make the connection explicit. These have now been spelled out.
Lines 506-509: please, give a sense to this statement in relation to this study. A sentence explaining the impact of Nox2 and Ros has been added.

Round 2
Reviewer 2 Report
Comments and Suggestions for Authors
Dear Authors, you extensively revised your manuscript according to my concerns/comments/suggestions. Now, it looks correct and precise. Best regards,
Reviewer 3 Report
Comments and Suggestions for Authors
Dear Authors, I recommend the acceptance of the manuscript. I just noted a mistake at line 560 in the L. helveticus name.